# Higher Diet Quality is Associated with Lower Odds of Low Hand Grip Strength in the Korean Elderly Population

**DOI:** 10.3390/nu11071487

**Published:** 2019-06-29

**Authors:** Hyesook Kim, Oran Kwon

**Affiliations:** Department of Nutritional Science and Food Management, Ewha Womans University, 52, Ewhayeodae-gil, Seodaemun-gu, Seoul 03760, Korea

**Keywords:** dietary quality, muscle strength, KNHANES

## Abstract

Single nutrients or food groups have been associated with physical performance. However, little is known about the association of overall diet quality with hand grip strength (HGS), a predictive parameter in the prognosis of chronic disease morbidity and mortality, or quality of life. This study examined the association between HGS and three indices—the Korean Healthy Eating Index (KHEI), the Alternate Mediterranean Diet (aMED), and Dietary Approaches to Stop Hypertension (DASH)—using data obtained on Korean elderly persons aged ≥65 years (*n* = 3675) from a nationally representative database. The cross-sectional data was collected as part of the Korea National Health and Nutrition Examination Survey (KNHANES, 2014–2016). Dietary intake data from the 24-h recall method were used to calculate diet quality scores. The cutoff value for low HGS was defined as the value corresponding to the lowest 20th percentile of HGS of the study population (men, 28.6 kg; women, 16.5 kg). Higher index scores for diet quality were associated with 32%–53% lower odds of low HGS. Better overall diet quality may be associated with higher HGS in the elderly Korean population.

## 1. Introduction

Aging is accompanied by widespread and typical changes in the human body. The most prominent physical change that occurs during the aging process is the loss of muscle mass, resulting in decreased muscle strength [1]. Reduced muscle strength is linked to a loss of independence and an increased risk of reduced mobility, disability, and mortality [2]. Muscle strength is one of the most decisive indicators of health status in older adults [3], and therefore, maintaining muscle strength is very important for reducing functional limitations in older adults.

Hand grip strength (HGS) is recognized as a general indicator of overall muscular strength [4], as well as an important biomarker of health and disease status [5]. HGS has been reported to be associated with various chronic diseases, including cardiovascular disease, diabetes, and metabolic syndrome [6,7,8,9]. Moreover, the predictive validity of HGS for decline in cognition, mobility, and mortality was established in a recent systematic review and meta-analysis [10].

HGS can be influenced by many factors, such as age [11], gender [12], race [13], and exercise [14]. Nutritional status is one of the major determinants of HGS with aging. Many studies have reported a link between HGS and intake of specific single nutrients, including protein [15], vitamin D [16], and antioxidants, like carotenoids [17], vitamin E [18], vitamin C [19], and selenium [20]. However, recent dietary research has focused on the whole diet, rather than single nutrients or individual food groups, because dietary components are consumed in combination and interact with one another. There is some evidence relating HGS to overall dietary quality indicators, such as the Healthy Eating Index (HEI) [15,21], Mediterranean Diet (MED) [22,23,24], Dietary Approaches to Stop Hypertension (DASH) [15], and Nordic Diet Score (NDS) [25], from European [22,23,24,25] and North American studies [15,21]. 

South Korea has one of the most rapidly aging populations in the world [26], and age-related poor muscle strength is becoming a major public health problem. Thus, more research on the association between overall diet quality and HGS, an important index of low muscle strength, is needed in the Korean population. While many studies have been conducted in Western countries, Jeong’s work [27] is the only research utilizing Korean subjects. Recently, Jeong et al. [27] reported on the positive relationship between the recommended food score (RFS), one of the indicators of overall diet quality, and HGS in elderly Korean women participating in the 2014–2015 National Fitness Award project. The Korean Ministry of Health and Welfare has begun to include the HGS measurement in the population-based Korea National Health and Nutrition Examination Survey (KNHANES) in 2014. Since then, research has been published on the relationship between HGS and some health outcomes, such as hypertension [28], chronic obstructive pulmonary disease [29], and quality of life [30,31]. However, to the best of our knowledge, no studies have reported on the relationship between overall diet quality and HGS among a representative Korean population.

Therefore, this study was conducted to investigate the association between diet quality indices—the Korean Healthy Eating Index (KHEI), the alternate Mediterranean Diet (aMED), and DASH—and HGS in the Korean elderly, using the nationally representative KNHANES data.

## 2. Materials and Methods 

### 2.1. Study Population

This study was based on data from the second and third years (2014–2015) of the KNHANES VI (2013–2015) and the first year (2016) of KNHANES VII (2016–2018), conducted by the Korea Centers for Disease Control and Prevention (KCDC). The KNHANES employs multi-stage stratified cluster sampling for the selection of household units among non-institutionalized residents in the Republic of Korea. The survey consists of a health interview, a health examination, and a nutrition assessment. This study was approved by the Institutional Review Board of the KCDC, and written informed consent was obtained from all subjects. Detailed information about the survey is available on the website [32] (http://knhanes.cdc.go.kr).

Among the participants, 4766 individuals (2077 men and 2689 women) aged ≥65 years participated in the survey. We excluded those with missing values for HGS (207 men and 509 women) and 24-h dietary recall (149 men and 167 women). Subjects (12 men and 47 women) with implausible energy intakes of <500 kcal or >8000 kcal were also excluded. The final sample for the analysis comprised of 3675 individuals (1709 men and 1966 women).

### 2.2. General Characteristics

The health interview and health examination portions of the survey were used to obtain the demographic and socioeconomic characteristics of the participants, including age, health status, body mass index (BMI), education level, smoking status, alcohol consumption, physical activity, and age at menarche.

Health status included physician-diagnosed chronic conditions (hypertension, hyperlipidemia, stroke, myocardial infarction or angina, arthritis, asthma, diabetes mellitus, cancer, depression, renal failure, and chronic obstructive pulmonary disease). BMI was calculated as weight (kg)/(height (m))^2^. Education level was categorized as elementary school or less versus more than elementary school. Smoking status was classified into three categories (non-smoker, ex-smoker, and current smoker). Alcohol consumption was classified into five categories (never, <1 drink/month, 1 drink/month, 2–4 drinks/month or ≥4 drinks/month within the last year). Physical activity was divided into five categories according to the number of days when strength exercise occurred within week prior (none, 1 day/week, 2 days/week, 3 days/week, 4 days/week or ≥5 days/week). For women, the age at menarche was recorded.

### 2.3. Dietary Assessment 

KNHANES conducts a 24-h recall survey for all participants aged over 12 months (1–80 years of age), whereas the food frequency questionnaire is conducted only for adults aged 19–64 years. Therefore, we used data from a single 24-h recall period to investigate the dietary intake of older adults aged over 65 years. Participants reported all foods and drinks consumed during the previous day in face-to-face interviews.

Table 1 identifies the components and standards for optimal scoring in the diet quality indices, such as KHEI, aMED, and DASH; specific details are described below.

#### 2.3.1. KHEI

The KHEI was developed [33] to quantify adherence to dietary guidelines for Korean adults [34], the 2010 Dietary Reference Intakes for Koreans [35], and the objectives of the Health Plan 2020 [36]. The KHEI scores 14 components for a maximum total of 100 points. Eight components—whole grains, refined grains, fruit intake (excluding juice), fruit intake (including juice), vegetable intake (excluding Kimchi and pickles), vegetable intake (including Kimchi or pickles), ratio of white meat-to-red meat, and percentages of energy from carbohydrates—are worth 0–5 points; six components—breakfast, milk and dairy, protein foods, sodium, percentages of energy from empty calorie foods, and percentages of energy from fat —are worth 0–10 points.

#### 2.3.2. aMED Score

The first Mediterranean diet score, developed by Trichopoulou et al. [37], takes into account the scientific literature on diet and chronic disease risk. The score we used, aMED, was modified by Fung et al. [38]. The aMED scores nine components for a total of 9 points. The intakes of nine food groups were dichotomized using sex-specific median values as the cutoff. A score of 1 was assigned for consumption above the median level of presumed beneficial foods (whole grains, vegetables (excluding potatoes), fruit (including juice), nuts, legumes, fish, and the ratio of monounsaturated fatty acids-to-saturated fatty acids) and for consumption below the median level of presumed detrimental foods (red and processed meat), and a score of 0 was assigned for all other. For alcohol, 1 point was assigned to men who consumed between 10 and 25 g/day and to women who consumed between 5 and 15 g/day, versus a score of 0.

#### 2.3.3. DASH Score

The DASH score was developed by Fung et al. [39] based on foods and nutrients emphasized or minimized in the DASH diet, according to an eating guide developed by the National Heart, Lung and Blood Institute [40]. DASH scores eight components (seven food groups and one nutrient)—each worth 5 points—for a maximum total of 40 points. The scoring system is based on sex-specific quintile rankings within the study population. Points range from 5 (highest quintile) to 1 (lowest quintile) for whole grains, vegetables (excluding potatoes), fruit (including juice), nuts and legumes, and low-fat dairy, and from 1 (highest quintile) to 5 (lowest quintile) for sugar-sweetened beverages, red and processed meat, and sodium. Among these eight components, the survey of low-fat dairy food intake was not performed in the 24-h recall portion of KNHANES. Therefore, in this study, low-fat dairy was used as a whole dairy product.

### 2.4. Hand Grip Strength

HGS was measured three times in each hand, using a digital hand dynamometer (digital grip strength dynamometer, TKK-5401, Takei Scientific Instruments Co., Ltd., Tokyo, Japan). Trained medical technicians instructed the seated subjects to hold the dynamometer with the second finger nodes of the working hand at an angle of 90° to the handle and to squeeze the handle as firmly as they could. After subjects slowly stood up, HGS was measured during expiration. Between each measurement, a 60 s resting interval was allowed. The HGS value used in the analysis was the highest of the six measured values [41].

### 2.5. Statistical analysis

All statistical analyses were performed using SAS software version 9.4 (SAS Institute, Cary, NC, USA). Due to the complex sampling design of the KNHANES study, the relevant primary sampling units, stratification, and sample weights were considered in our analysis. For descriptive statistics, continuous variables are expressed as weighted means and SEMs, and categorical variables are expressed as numbers and weighted percentages, using the procedures of SURVEYMEANS and SURVEYFREQ, respectively. SURVEY LOGISTIC analysis was performed to estimate the odds ratios (ORs) and 95% confidence intervals (CIs) for risk of low muscle strength across tertiles of each diet quality index, with the 1st tertile set as the reference. Model 1 was adjusted for age (years), number of physician-diagnosed chronic conditions (n), BMI (kg/m^2^), education (≤elementary school, >elementary school), smoking (non-smoker, ex-smoker, or current smoker), alcohol consumption (never, <1 drink/month, 1 drink/month, 2–4 drinks/month or ≥4 drinks/month), energy intake (kcal), and age at menarche (women only) (years). Model 2 was adjusted for the variables in model 1, plus physical activity assessed by days of strength exercise within the last week (never, 1 day/week, 2 days/week, 3 days/week, 4 days/week or ≥5 days/week). All reported probability tests were two-sided, and differences were considered significant at the 5% level.

## 3. Results

### 3.1. General Characteristics

Low muscle strength was defined as the lower 20th percentile of HGS (men, 28.6 kg; women, 16.4 kg), and 344 men (19.5%) and 388 women (19.7%) were classified into the low muscle strength group. Table 2 shows the characteristics of the men and women in the highest tertile (3rd tertile; most optimal diet quality) compared with the lowest tertile (1st tertile; poorest diet quality) for each diet quality index. Across all indices, both men and women in the 3rd tertile were more likely to be more educated, never to have smoked, be more physically active, and consume dietary supplements. Across all indices, men in the 3rd tertile had a high BMI; in contrast, there was no difference in the BMI across the tertile groups among women. Across all indices, women in the 3rd tertile were more likely to be younger, have higher energy intakes, and a younger age at menarche. For DASH, men in the 3rd tertile had a higher number of physician-diagnosed chronic conditions; in contrast, women in the 3rd tertile had a lower number of physician-diagnosed chronic conditions. 

### 3.2. Association of Diet Quality with Grip Strength

Correlations among the total scores for all pairs of indices are presented in Table 3. For men, the correlations ranged from 0.38 (for KHEI and aMED) to 0.59 (for aMED and DASH). For women, the correlations ranged from 0.43 (for KHEI and aMED) to 0.60 (for aMED and DASH). All correlations were significant (*p* < 0.0001).

Table 4 shows that across all indices, men and women in the 3rd tertile, compared with men and women in the 1st tertile had 32–53% lower odds of low HGS. For both men and women, when comparing the 3rd and 1st tertile, the direction and magnitude of the adjusted ORs consistently indicated an inverse association for risk of low HGS. For example, the ORs for risk of low HGS were as follows: KHEI (OR: 0.57, 95% CI: 0.35–0.91, *p*-trend = 0.015), aMED (OR: 0.64, 95% CI: 0.44–0.93, *p*-trend = 0.014), and DASH (OR: 0.63, 95% CI: 0.41–0.94, *p*-trend = 0.032) in men; and KHEI (OR: 0.68, 95% CI: 0.46–0.99, *p*-trend = 0.053), aMED (OR: 0.47, 95% CI: 0.31–0.69, *p*-trend < 0.001), and DASH (OR: 0.52, 95% CI: 0.34–0.79, *p*-trend = 0.004) in women.

## 4. Discussion

In this nationally representative cross-sectional study, we found that higher index scores for diet quality were associated with 32%–53% lower odds of low HGS in elderly Korean men and women, which were similar across all three diet quality indices—KHEI, aMED, and DASH. To our knowledge, this is the first study to identify that better overall diet quality may be associated with higher HGS in the elderly Korean population. Moreover, no previous literature studies have compared these specific indices within the same Korean population for the prevalence of low HGS.

The positive associations between diet quality and HGS are consistent with other recent (non-Korean) cross-sectional studies [15,24]. Fanelli et al. [15] found that adherence to the DASH eating pattern was positively associated with relative HGS in men and women in the USA. Barrea et al. [24] reported a positive association between adherence to the MED and HGS in a sample of active Italian senior women, stratified according to the HGS > cutoff-point of 20 kg. Conversely, a cross-sectional study conducted in Italian elderly men and women [23] found no statistically significant associations between higher adherence to a MED and HGS, and a longitudinal study conducted in Canada [21] showed no significant association between diet quality measured using the Canadian Healthy Eating Index at baseline and the maintenance of three measures of muscle strength, including HGS.

Some studies [22,42] have shown that high adherence to a MED is significantly related to frailty but not to HGS, one of the criteria included in the definition of frailty. Bollwein et al. [42] found an association between a healthy diet, as measured using a MED score, and lower odds of frailty (defined as having three or more of the following five criteria: weight loss, feelings of exhaustion, poor HGS, low walking speed, or low physical activity). The association existed for low walking speed and low physical activity but not exhaustion and low HGS in elderly German individuals aged 75 years or older. Talegawkar et al. [22] reported that high adherence to a MED at baseline was associated with a lower risk of developing frailty (defined as having two or more of the following four criteria: feelings of exhaustion, poor HGS, low walking speed, or low physical activity) during a 6-year follow-up in a cohort of older, community-dwelling Italian men and women aged 65 years or older. However, when the authors looked at a high adherence to a MED and the relevance of each factor to frailty, diet was significantly related to low walking speed and low physical activity but not to feelings of exhaustion or poor HGS [22]. Therefore, the current evidence regarding the relationship between diet quality and HGS is inconsistent. 

Some studies have examined the relationship between diet quality and leg strength (knee extension), instead of HGS, as an indicator of muscle strength. A cross-sectional study of men and women aged 60 years or older in the USA [43] found that a higher total HEI 2005 score was associated with greater knee extension strength; however, this association was not statistically significant after adjustment for physical activity. Similarly, a longitudinal study of elderly Japanese women [44] found no significant association between diet quality (dietary variety) and knee extension strength. In contrast, a longitudinal study of Finnish elderly [25] showed that adherence to the NDS predicted greater HGS and knee extension strength 10 years later among women but not among men. The results of the study on leg strength and diet quality are also inconsistent. Therefore, a comprehensive study is needed in the future.

In our study, the exact mechanism of the relationship between all three diet quality indices—KHEI, aMED, and DASH—and HGS is not known, but several possible links could mediate the positive relationships between them. For example, elderly people with healthy dietary patterns are more likely to also have a healthy lifestyle, characteristics of which include participation in physical activity and regular exercise [45]. Such activity may help prevent age-related decreases in muscle mass or function [14,46]. Moreover, adherence to a high-quality diet, quantified using indices such as HEI, aMED, and DASH, has been shown to decrease the likelihood of obesity [47,48,49], a factor closely associated with decline in muscle strength and mass [50]. In addition, higher diet quality seems to be associated with antioxidant and anti-inflammatory responses. Adherence to KHEI, aMED, and DASH eating patterns, which are rich in antioxidant nutrients such as magnesium and vitamins C and E, were associated with lower blood levels of inflammatory markers, such as C-reactive protein and interleukin-6 [51]. In particular, a high score on the aMED indicates a higher intake of omega-3 fatty acids and monounsaturated fatty acids, which appear to be associated with anti-inflammatory effects [52]. Chronic low-grade systemic inflammation and oxidative stress can generate catabolism and increase protein turnover in skeletal muscle, reducing strength [53,54], and can also increase the generation of reactive oxygen species, resulting in an overload of the antioxidant defense system [55].

We observed that across all indices, both men and women in the 3rd tertile were more likely to be more educated and physically active, have never smoked, and consume dietary supplements. This is consistent with previous studies [56,57,58] showing that people with higher diet quality are more likely to have higher levels of education and healthier lifestyles than those who have lower diet quality. We also observed that across all indices, men in the 3rd tertile had higher BMIs, compared to those in the lowest tertile, and women in the 3rd tertile had higher energy intakes when compared to those in the lowest tertile. Given that older adults are at an increased susceptibility of malnutrition due to age-associated changes in metabolism and physiology [59], our results demonstrate that increased BMI and energy intake may be favorable for the elderly.

In the present study, the relationship between diet quality indicators and HGS existed in both men and women. Similarly, a study of the elderly in the USA showed a relationship between diet quality and HGS in both men and women [15]. Some studies [25,60,61] have indicated that the effects of diet quality index or dietary pattern on HGS might be different for men and women; however, the evidence was inconsistent. One of the cross-sectional studies [60] found that a healthier eating pattern was independently associated with higher HGS in women but not in men. The longitudinal study conducted on elderly Finnish people [25] reported that the healthy NDS predicts higher HGS 10 years later in senior women but not senior men. In a longitudinal study conducted in the UK [61], dietary patterns high in red meats, potato and gravy, or butter were associated with a lower HGS and a greater decline in HGS in men, whereas no association was observed in women. Since no other research on this topic has been conducted on Koreans, it is not possible to know whether the gender difference is dependent upon race. Further research is needed in the future.

One important limitation of this study was the inability to define causal associations with this dataset because of the cross-sectional study design. Further prospective studies are warranted to investigate the causal associations between diet quality or dietary pattern and HGS. Another limitation is that our dietary data was derived from a single 24-h dietary recall survey, which may provide an inaccurate estimate of a normal diet. However, the KNHANES report has shown that variations in data (3.9% for energy, within 10% for macro- and micro-nutrients) from a single day and 2–10 days of 24-h dietary recall were not significantly different [62]. Nevertheless, to the best of our knowledge, this is the first study identifying that better overall diet quality may be associated with higher HGS in the elderly Korean population, based on a nationally representative sample. This study has the potential to inform both policymakers and persons developing dietary guidelines about the role of diet quality or dietary patterns on the health of older adults.

## 5. Conclusions

We found that better overall diet quality may be associated with higher HGS in the elderly Korean population. Understanding the relationship between diet quality and HGS may offer opportunities for prevention and timely interventions. Future studies with larger sample sizes and prospective or interventional design are needed to further improve our knowledge about the association between diet quality or dietary patterns and HGS.

## Figures and Tables

**Table 1 nutrients-11-01487-t001:** Scoring standards for each component of the KHEI, aMED, and DASH score ^1.^

	KHEI ^2^			aMED ^3^			DASH ^4^		
Component	Maximum Score	Criteria for Maximum Score	Criteria for Minimum Score	Maximum Score	Criteria for Maximum Score	Criteria for Minimum Score	Maximum Score	Criteria for Maximum Score	Criteria for Minimum Score
Breakfast	10	Have breakfast in 2 d	Nothing	—	—	—	—	—	—
Whole grains	5	≥ 1 svg/d	0 svg/d	1	Median or greater	Less than median	5	Highest quintile	Lowest quintile
Refined grains	5	Male: ≤ 4 svg/d, Female: ≤ 3 svg/d	Male: ≥ 5 svg/d, Female: ≥ 4 svg/d	—	—	—	—	—	—
Nuts and legumes							5	Highest quintile	Lowest quintile
Nuts	—	—	—	1	Median or greater	Less than median	—	—	—
Legumes	—	—	—	1	Median or greater	Less than median	—	—	—
Fruits including juice	5	Male: ≥ 3 svg/d, Female: ≥ 2 svg/d	0 svg/d	1	Median or greater	Less than median	5	Highest quintile	Lowest quintile
Fruit excluding juice	5	Male: ≥ 1.5 svg/d, Female: ≥ 1 svg/d	0 svg/d	—	—	—	—	—	—
Vegetables including Kimchi or pickles	5	≥ 7 svg/d	0 svg/d	1	Median or greater	Less than median	5	Highest quintile	Lowest quintile
Vegetables excluding Kimchi and pickles	5	≥ 4 svg/d	0 svg/d	—	—	—	—	—	—
Milk and dairy	10	≥ 1 svg/d	0 svg/d	—	—	—	5	Highest quintile	Lowest quintile
Protein foods	10	Male: ≥ 5 svg/d, Female: ≥ 4 svg/d	0 svg/d	—	—	—	—	—	—
Red and processed meat	—	—	—	1	Median or greater	Less than median	5	Lowest quintile	Highest quintile
Ratio of white meat to red meat	5	4: 1 (≥ 4)	0 svg/d (Red meat only)	—	—	—	—	—	—
Fish	—	—	—	1	Median or greater	Less than median	—	—	—
Sugar-sweetened beverages	—	—	—	—	—	—	5	Lowest quintile	Highest quintile
Alcohol	—	—	—	1	Male: 10-25 g/d Female: 5-15 g/d	Male: < 10 or > 25 g/d Female: < 5 or > 15 g/d	—	—	—
Sodium	10	≤ 2000 mg/d	> 85 percentile	—	—	—	5	Lowest quintile	Highest quintile
Percentages of energy from empty calorie foods	10	≤ 5% energy	≥ 10% energy	—	—	—	—	—	—
Percentages of energy from carbohydrates	5	55~70%	< 15 percentile > 85 percentile	—	—	—	—	—	—
Percentages of energy from fat	10	15~25%	< 15 percentile > 85 percentile	—	—	—	—	—	—
MUFA: SFA ratio	—	—	—	1	Median or greater	Less than median	—	—	—

^1^ Scoring standards are based on one serving size from the Food nutritional value table easy to understand by consumers in one serving by National Academy of Agricultural Science ; KHEI, Korean Healthy Eating Index; aMED, alternate Mediterranean Diet; DASH, Dietary Approaches to Stop Hypertension; svg, serving; d, day; —, not applicable; MUFA, Monounsaturated Fatty Acid; SFA, Saturated Fatty Acid. ^2^ KHEI: 100 points total; 14 components: 5–10 points each. ^3^ aMED: 9 points total; 9 components: 1 point each. ^4^ DASH: 40 points total; 8 components: 5 points each.

**Table 2 nutrients-11-01487-t002:** Descriptive characteristics of elderly men and women in the 2014-2016 KNHANES based on tertiles of the KHEI, aMED, and DASH score ^1.^

	KHEI						aMED						DASH					
	T1	T2	T3	*p*-Value	*p*-Value (T1 vs. T3)	*p*-Value (T2 vs. T3)	T1	T2	T3	*p*-Value	*p*-Value (T1 vs. T3)	*p*-Value (T2 vs. T3)	T1	T2	T3	*p*-Value	*p*-Value (T1 vs. T3)	*p*-Value (T2 vs. T3)
**Men**																		
*n*	569	570	570				632	373	704				594	489	626			
Low muscle strength ^2^, n (%)	143 (24.0)	121 (21.5)	80 (13.0)	<0.001			174 (27.4)	75 (20.4)	95 (12.5)	<0.001			131 (21.7)	111 (22.0)	102 (15.5)	0.018		
Age, y	72.9 ± 0.2	72.1 ± 0.2	72.0 ± 0.2	0.007	0.004	0.689	73.3 ± 0.2	72.4 ± 0.3	71.5 ± 0.2	<0.001	<0.001	0.012	72.2 ± 0.2	72.8 ± 0.3	72.1 ± 0.2	0.080	0.582	0.030
BMI, kg/m^2^	23.3 ± 0.2	23.8 ± 0.1	23.9 ± 0.1	0.008	0.005	0.907	23.4 ± 0.2	23.8 ± 0.2	23.8 ± 0.1	0.036	0.015	0.971	23.5 ± 0.2	23.5 ± 0.1	24.0 ± 0.1	0.038	0.028	0.030
Education, *n* (%)																		
<= Elementary school	284 (56.1)	220 (42.7)	141(26.6)	<0.001			293 (53.9)	141 (39.3)	211 (32.4)	<0.001			261 (49.5)	188 (42.7)	196 (33.5)	<0.001		
> Elementary school	219 (43.9)	308 (57.3)	392 (73.4)				271 (46.1)	207 (60.7)	441 (67.6)				272 (50.5)	254 (57.3)	393 (66.5)			
No response	66	42	37				68	25	52				61	47	37			
Smoking status, *n* (%)																		
Nonsmoker	96 (18.0)	96 (16.0)	139 (27.1)	<0.001			113 (19.7)	68 (17.6)	150 (22.3)	0.060			93 (16.4)	99 (20.5)	139 (23.8)	<0.001		
Ex-smoker	298 (58.5)	343 (63.3)	340 (59.7)				333 (57.3)	228 (63.9)	420 (61.6)				329 (59.2)	271 (59.0)	381 (63.2)			
Current smoker	122 (23.6)	111 (20.7)	69 (13.2)				135 (23.1)	61 (18.5)	106 (16.1)				139 (24.4)	86 (20.5)	77 (13.0)			
No response	53	20	22				51	16	28				33	33	29			
Alcohol consumption, *n* (%)																		
Never	154 (30.3)	189 (32.4)	201 (37.6)	<0.001			218 (39.0)	109 (30.1)	217 (30.7)	0.057			187 (33.4)	145 (31.3)	212 (35.2)	0.362		
< 1 Months	57 (10.4)	83 (15.9)	110 (19.7)				87 (14.4)	59 (16.3)	104 (15.9)				80 (13.3)	71 (16.2)	99 (16.9)			
1 Months	75 (13.4)	108 (19.8)	114 (20.2)				90 (15.1)	78 (22.0)	129 (18.1)				93 (17.4)	88 (17.5)	116 (18.8)			
2-4 Months	96 (19.8)	89 (17.0)	76 (13.8)				81 (14.5)	53 (15.5)	127 (19.4)				93 (17.1)	81 (19.7)	87 (14.3)			
>= 4 Months	136 (26.1)	82 (14.8)	47 (8.7)				105 (17.0)	59 (16.1)	101 (15.8)				110 (18.8)	72 (15.3)	83 (14.8)			
No response	51	19	22				51	15	26				31	32	29			
Days of strength exercise (days/week), *n* (%)																		
No	403 (79.6)	363 (66.4)	347 (65.9)	<0.001			449 (78.8)	255 (73.2)	409 (61.9)	<0.001			415 (76.7)	330 (73.9)	368 (62.1)	<0.001		
1 days/week	6 (1.1)	11 (2.0)	6 (0.9)				11 (1.7)	1 (0.3)	11 (1.6)				8 (1.7)	9 (1.8)	6 (0.7)			
2 days/week	15 (3.3)	25 (4.5)	27 (5.7)				17 (3.2)	14 (3.5)	36 (6.1)				21 (3.8)	15 (3.9)	31 (5.6)			
3 days/week	16 (3.2)	20 (4.2)	24 (4.1)				15 (2.8)	15 (4.1)	30 (4.5)				16 (3.0)	13 (2.7)	31 (5.5)			
4 days/week	8 (1.3)	8 (1.8)	20 (3.9)				8 (1.3)	11 (3.5)	17 (2.6)				10 (2.1)	8 (2.0)	18 (2.9)			
>=5 days/week	58 (11.4)	101 (21.2)	109 (19.5)				65 (12.1)	54 (15.4)	149 (23.2)				65 (12.8)	68 (15.8)	135 (23.2)			
No response	63	42	37				67	23	52				59	46	37			
Dietary supplement use, n (%)																		
No	385 (68.2)	342 (59.0)	295 (49.8)	<0.001			433 (69.1)	218 (57.2)	371 (51.5)	<0.001			392 (65.2)	296 (60.1)	334 (52.4)	<0.001		
Yes	184 (31.8)	228 (41.0)	275 (50.2)				199 (30.9)	155 (42.8)	333 (48.5)				202 (34.8)	193 (39.9)	292 (47.6)			
Energy intake, kcal/d	1916.4 ± 37.3	1969.8 ± 35.1	2002.8 ± 33.67	0.209	0.078	0.482	1746.6 ± 32.5	1934.3 ± 41.3	2161.4 ± 30.2	<0.001	<0.001	<0.001	1917.4 ± 33.8	1981.7 ± 42.6	1991.4 ± 32.1	0.222	0.105	0.856
Number of physician diagnosed chronic conditions	1.3 ± 0.1	1.3 ± 0.1	1.4 ± 0.1	0.407	0.246	0.232	1.2 ± 0.1	1.4 ± 0.1	1.3 ± 0.1	0.132	0.103	0.607	1.2 ± 0.1	1.2 ± 0.1	1.4 ± 0.1	0.028	0.016	0.039
**Women**																		
*n*	655	656	655				736	410	820				633	694	639			
Low muscle strength ^2^, n (%)	171 (26.2)	130 (19.3)	87 (14.0)	<0.001			200 (27.3)	86 (20.5)	102 (12.6)	<0.001			158 (24.4)	151 (22.4)	79 (12.5)	<0.001		
Age, y	74.0 ± 0.2	72.8 ± 0.2	71.5 ± 0.2	<0.001	<0.001	<0.001	74.0 ± 0.2	73.1 ± 0.3	71.5 ± 0.2	<0.001	<0.001	<0.001	73.4 ± 0.2	73.1 ± 0.2	71.8 ± 0.2	<0.001	<0.001	<0.001
BMI, kg/m^2^	24.4 ± 0.2	24.4 ± 0.2	24.5 ± 0.1	0.779	0.492	0.643	24.4 ± 0.1	24.5 ± 0.2	24.4 ± 0.1	0.965	0.896	0.874	24.4 ± 0.2	24.4 ± 0.2	24.5 ± 0.1	0.936	0.718	0.908
Education, n (%)																		
<= Elementary school	517 (87.8)	464 (76.9)	361 (58.6)	<0.001			547 (83.3)	295 (78.7)	500 (63.8)	<0.001			478 (83.9)	465 (72.7)	399 (66.1)	<0.001		
> Elementary school	58 (12.2)	135 (23.1)	249 (41.4)				100 (16.7)	76 (21.3)	266 (36.2)				86 (16.1)	160 (27.3)	196 (33.9)			
No response	80	57	45				89	39	54				69	69	44			
Smoking status, n (%)																		
Nonsmoker	537 (89.8)	579 (94.6)	600 (95.4)	0.020			609 (90.3)	368 (94.5)	739 (95.4)	0.023			524 (89.8)	602 (93.9)	590 (96.1)	0.003		
Ex-smoker	34 (5.4)	19 (3.4)	18 (2.9)				33 (4.9)	10 (3.4)	28 (3.1)				29 (5.0)	27 (4.2)	15 (2.4)			
Current smoker	20 (4.8)	11 (2.1)	12 (1.8)				24 (4.8)	6 (2.1)	13 (1.5)				22 (5.2)	12 (1.9)	9 (1.5)			
No response	64	47	25				70	26	40				58	53	25			
Alcohol consumption, n (%)																		
Never	380 (62.6)	387 (64.0)	373 (57.8)	<0.001			439 (65.0)	241 (62.8)	460 (57.7)	0.018			361 (60.8)	396 (61.1)	383 (62.3)	0.145		
< 1 drink/month	132 (22.7)	169 (27.7)	189 (30.5)				149 (23.1)	106 (26.9)	235 (30.7)				143 (25.6)	173 (27.5)	174 (28.2)			
1 drink/month	40 (7.0)	33 (4.9)	44 (7.5)				43 (6.6)	24 (6.0)	50 (6.5)				38 (6.7)	42 (6.8)	37 (5.9)			
2-4 drink/month	20 (3.7)	18 (2.9)	19 (3.2)				19 (2.5)	9 (2.4)	29 (4.3)				19(3.4)	20 (3.4)	18 (2.9)			
>= 4 drink/month	22 (4.1)	4 (0.4)	5 (1.0)				18 (2.8)	7 (1.9)	6 (0.8)				18 (3.4)	9 (1.2)	4 (0.8)			
No response	61	45	25				68	23	40				54	54	23			
Days of strength exercise (days/week), n (%)																		
No	542 (93.5)	546 (92.8)	516 (83.9)	<0.001			608 (93.2)	338 (91.9)	658 (86.1)	0.002			528 (93.4)	565 (90.4)	511 (86.2)	0.009		
1 day/week	4 (1.0)	7 (1.0)	13 (2.0)				9 (1.4)	3 (0.7)	12 (1.5)				9 (1.6)	9 (1.2)	6 (1.1)			
2 days/week	9 (1.5)	11 (1.9)	9 (1.4)				7 (1.2)	6 (1.1)	16 (2.1)				9 (1.5)	10 (1.6)	10 (1.6)			
3 days/week	3 (0.4)	11 (1.5)	19 (3.2)				6 (0.7)	3 (0.7)	24 (3.1)				5 (0.6)	9 (1.4)	19 (3.1)			
4 days/week	4 (0.7)	2 (0.3)	14 (2.3)				4 (0.6)	4 (0.8)	12 (1.6)				3 (0.4)	9 (1.4)	8 (1.4)			
>=5 days/week	16 (3.0)	21 (2.8)	42 (7.3)				16 (2.9)	17 (4.8)	46 (5.5)				12 (2.5)	24 (4.0)	43 (6.6)			
No response	77	58	42				86	39	52				67	68	42			
Dietary supplement use, *n* (%)																		
No	393 (57.8)	338 (50.3)	256 (37.9)	<0.001			441 (58.3)	196 (43.5)	350 (42.3)	<0.001			362 (55.9)	348 (48.5)	277 (41.4)	<0.001		
Yes	262 (42.2)	318 (49.7)	399 (62.1)				295 (41.7)	214 (56.5)	470 (57.7)				271 (44.1)	346 (51.5)	362 (58.6)			
Energy intake, kcal/d	1438.4 ± 30.6	1454.8 ± 26.6	1537.1 ± 25.3	0.015	0.012	0.018	1293.0 ± 24.1	1422.6 ± 27.3	1669.5 ± 27.9	<0.001	<0.001	<0.001	1429.3 ± 28.2	1471.9 ± 27.2	1529.4 ± 28.5	0.043	0.013	0.130
Number of physician diagnosed chronic conditions	1.7 ± 0.1	1.8 ± 0.1	1.9 ± 0.1	0.096	0.042	0.662	1.8 ± 0.1	1.8 ± 0.1	1.8 ± 0.1	0.825	0.946	0.554	1.7 ± 0.1	1.8 ± 0.1	1.9 ± 0.1	0.038	0.011	0.166
Age at menarche, y	16.1 ± 0.1	15.5 ± 0.1	15.4 ± 0.1	<0.001	<.001	0.418	15.9 ± 0.1	15.8 ± 0.1	15.5 ± 0.1	0.005	0.002	0.018	16.0 ± 0.1	15.7 ± 0.1	15.4 ± 0.1	<0.001	<0.001	0.007

^1^ Values are means ± SDs unless otherwise specified. KNHANES, Korean National Health and Nutrition Examination Survey; KHEI, Korean Healthy Eating Index; aMED, alternate Mediterranean Diet; DASH, Dietary Approaches to Stop Hypertension; T, tertile, y, year. ^2^ Low muscle strength was defined as the lower 20th percentile of hand grip strength (men, 28.6 kg; women, 16.4 kg).

**Table 3 nutrients-11-01487-t003:** Correlation coefficients among diet quality indices for the KHEI, aMED, and DASH score of elderly men and women in the 2014–2016 KNAHNES ^1.^

	KHEI		aMED		DASH	
	Men	Women	Men	Women	Men	Women
KHEI	1.00	1.00	0.38	0.43	0.53	0.55
aMDS	0.38	0.43	1.00	1.00	0.59	0.60
DASH	0.53	0.55	0.59	0.60	1.00	1.00

^1^ All *p* < 0.0001. KNHANES, Korean National Health and Nutrition Examination Survey; KHEI, Korean Healthy Eating Index; aMED, alternate Mediterranean Diet; DASH, Dietary Approaches to Stop Hypertension.

**Table 4 nutrients-11-01487-t004:** Multivariate odds ratios (ORs) and 95% confidence intervals (CIs) for low muscle strength ^1^ according to tertiles of diet quality indices for the KHEI, aMED, and DASH score among elderly men and women in the 2014–2016.

	Men						Women					
	Unadjusted		Model 1		Model 2		Unadjusted		Model 1		Model 2	
	OR	(95% CI)	OR	(95% CI)	OR	(95% CI)	OR	(95% CI)	OR	(95% CI)	OR	(95% CI)
KHEI												
T1	1.00 (ref)		1.00 (ref)		1.00 (ref)		1.00 (ref)		1.00 (ref)		1.00 (ref)	
T2	0.87	(0.637,1.178)	1.12	(0.829,1.730)	1.27	(0.878,1.837)	0.67	(0.504,0.899)	0.77	(0.536,1.106)	0.75	(0.522,1.074)
T3	0.47	(0.326,0.681)	0.55	(0.344,0.875)	0.57	(0.357,0.919)	0.46	(0.332,0.634)	0.66	(0.456,0.967)	0.68	(0.468,0.999)
P for trend	<0.001		0.008		0.015		<0.001		0.027		0.053	
aMED												
T1	1.00 (ref)		1.00 (ref)		1.00 (ref)		1.00 (ref)		1.00 (ref)		1.00 (ref)	
T2	0.68	(0.476,0.968)	0.89	(0.594,1.343)	0.91	(0.602,1.377)	0.69	(0.486,0.972)	0.74	(0.496,1.102)	0.74	(0.499,1.110)
T3	0.38	(0.275,0.517)	0.59	(0.409,0.861)	0.64	(0.440,0.933)	0.38	(0.280,0.526)	0.45	(0.308,0.668)	0.47	(0.319,0.699)
P for trend	<0.001		0.005		0.014		<0.001		<0.001		<0.001	
DASH												
T1	1.00 (ref)		1.00 (ref)		1.00 (ref)		1.00 (ref)		1.00 (ref)		1.00 (ref)	
T2	0.88	(0.658,1.186)	0.78	(0.545,1.119)	0.81	(0.560,1.158)	0.84	(0.618,1.153)	0.80	(0.546,1.176)	0.82	(0.884,1.214)
T3	0.66	(0.469,0.931)	0.57	(0.377,0.859)	0.63	(0.413,0.948)	0.54	(0.386,0.763)	0.51	(0.334,0.769)	0.52	(0.343,0.796)
P for trend	0.018		0.010		0.032		<0.001		0.002		0.004	

^1^ Low muscle strength was defined as the lower 20th percentile of hand grip strength (men, 28.6 kg; women, 16.4 kg). KNHANES, Korean National Health and Nutrition Examination Survey; KHEI, Korean Healthy Eating Index; aMED, alternate Mediterranean Diet; DASH, Dietary Approaches to Stop Hypertension; T, tertile; ref, reference. Model 1: adjusted for age, BMI, education, smoking, alcohol drinking, number of physician diagnosed chronic conditions, energy intake, and age at menarche (women only) Model 2: adjusted for the variables in model 1, and physical activity

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
