# Peer review of "Higher Diet Quality is Associated with Lower Odds of Low Hand Grip Strength in the Korean Elderly Population"

_nutrients, 2019, doi:10.3390/nu11071487_

Round 1
Reviewer 1 Report
This study examined the association between handgrip strength (HGS) and three indices—the Korean Healthy Eating Index (KHEI), the alternate Mediterranean Diet (aMED), and Dietary Approaches to Stop Hypertension (DASH)—based on the nationwide database in Korea elderly aged ≥ 65 years. Authors suggested that higher index scores for diet quality were associated with a 32–53% lower odds of low HGS. A better overall diet quality may be associated with a higher HGS in the elderly Korean population.
This manuscript is written well, so I recommend to publish this after revision.
HGS is predictor of prognosis not only apparently healthy population but cardiac diseases, so I reccomend to add these references on introduction of this manuscript as a references.
1)
Handgrip strength as a predictor of prognosis in Japanese patients with congestive heart failure.
Izawa KP et al,
Eur J Cardiovasc Prev Rehabil. 2009 Feb;16(1):21-7.
2)
Grip strength predicts cardiac adverse events in patients with cardiac disorders: an individual patient pooled meta-analysis.
Pavasini R, et al.
Heart. 2019 Jun;105(11):834-841.
Author Response
Reviewer's Comments
This study examined the association between handgrip strength (HGS) and three indices—the Korean Healthy Eating Index (KHEI), the alternate Mediterranean Diet (aMED), and Dietary Approaches to Stop Hypertension (DASH)—based on the nationwide database in Korea elderly aged ≥ 65 years. Authors suggested that higher index scores for diet quality were associated with a 32–53% lower odds of low HGS. A better overall diet quality may be associated with a higher HGS in the elderly Korean population.
This manuscript is written well, so I recommend to publish this after revision.
Authors: We sincerely appreciate your kind comments. Thank you again for taking the time to review this paper. Please see our detailed responses below.
HGS is predictor of prognosis not only apparently healthy population but cardiac diseases, so I reccomend to add these references on introduction of this manuscript as a references.
1) Handgrip strength as a predictor of prognosis in Japanese patients with congestive heart failure. Izawa KP et al, Eur J Cardiovasc Prev Rehabil. 2009 Feb;16(1):21-7.
2) Grip strength predicts cardiac adverse events in patients with cardiac disorders: an individual patient pooled meta-analysis. Pavasini R, et al. Heart. 2019 Jun;105(11):834-841.
Authors: Thank you for your suggestion. We agree with this comment and added these references in the introduction (Lines 35 & 321-328).
Reviewer 2 Report
The manuscript 'Higher diet quality is associated with lower odds of low hand grip strength in Korean elderly' by Kim and Kwon describes the relation between different dietary patterns and handgrip strength. This topic is of high interest, as the approach to investigate dietary patterns instead of single nutrients or food groups is an up-to-date method to analyze nutritional quality.
However, before being ready for publication, some points need to be considered.
Major:
- There is no discussion about descriptive characteristics of the study, such as about age, BMI, energy intake,... This should be added to the discussion.
- In general, the discussion needs more depth/details. e.g.: discuss/compare the different dietary indices
- also, in Table 2, for men, the percentage of doing strength training on 5 or more days per week is surprisingly high with 11.4% (T1), 21.2 % (T2) and 19.5% (T3). please explain.
- I would expect to see more speculations about why the quality of the diet might be linked to better HGS. there must be data/reports about connections between diet, lifestyle, physical activity and strength parameters. The discussion and paper would benefit from information about possible links.
Minor:
- line 17: delete the word "of"
- line 29: "...increased risk of mobility..." is the wrong term; should probably be 'morbidity' or 'reduced mobility'
- line 81 & 151: use superscript for the BMI - kg/m2
- line 212: "...high adherence to a MED is significantly related to frailty..." is contradictory to what is written in the same paragraph in the next lines - please revise.
- line 246 & 250: the reference 22 seems to be wrong in both lines - should be reference 23 in my opinion. Please re-check all references!
Author Response
Reviewer's Comments
The manuscript 'Higher diet quality is associated with lower odds of low hand grip strength in Korean elderly' by Kim and Kwon describes the relation between different dietary patterns and handgrip strength. This topic is of high interest, as the approach to investigate dietary patterns instead of single nutrients or food groups is an up-to-date method to analyze nutritional quality.
However, before being ready for publication, some points need to be considered.
Authors: The critical comments and helpful suggestions are most appreciated. Please find the responses to the detailed comments below.
Major:
- There is no discussion about descriptive characteristics of the study, such as about age, BMI, energy intake,... This should be added to the discussion. In general, the discussion needs more depth/details. e.g.: discuss/compare the different dietary indices
Authors: Thank you for your suggestion. Based on your suggestion, we have added discussion about descriptive characteristics of the study, including other diet-related indicators such as energy and supplement intake (Lines 261-268).
- also, in Table 2, for men, the percentage of doing strength training on 5 or more days per week is surprisingly high with 11.4% (T1), 21.2 % (T2) and 19.5% (T3). please explain.
Authors: The questions related to strength exercises are as follows: [In the past week, how many days did you do your strength exercises such as push-ups, sit-ups, dumbbells, weights, and metal bars]. Table 2 shows that the percent of people who do not do strength exercise is about 70%, and the percentage of people who do it more than five times per week is much higher than those who strength exercise one, two, three, or four times per week. This means that the percentage of elderly who do not strength exercise is highest, but among individuals who do strength exercise, more frequent strength exercise is more common than less frequent. According to Korean health statistics from 2016, the percent of people who did strength exercise more than two days per week was 29.6% and 23.2% in men 60-69 years old and 70-80 years old, respectively. The corresponding values for women were 15.0% and 5.5%, respectively.
- I would expect to see more speculations about why the quality of the diet might be linked to better HGS. there must be data/reports about connections between diet, lifestyle, physical activity and strength parameters. The discussion and paper would benefit from information about possible links.
Authors: The critical comments are most appreciated. Based on your suggestion, we have supplemented the related paragraphs to include more speculation about why diet quality might be linked to increased HGS (Lines 248-251).
Minor:
- line 17: delete the word "of"
Authors: Thank you for finding this typo; we have made the correction (Line 17).
- line 29: "...increased risk of mobility..." is the wrong term; should probably be 'morbidity' or 'reduced mobility'
Authors: The critical comments are most appreciated. We revised it to ‘reduced mobility’ (Line 29).
- line 81 & 151: use superscript for the BMI - kg/m2
Authors: This has been revised (Line 83 & 162).
- line 212: "...high adherence to a MED is significantly related to frailty..." is contradictory to what is written in the same paragraph in the next lines - please revise.
Authors: We have revised this entire paragraph to include more detail to ensure that there is no misunderstanding. In other words, high adherence to a MED was associated with frailty, but when researchers analyzed the relevance of each of the criteria contained in the frailty definition, data indicated that MED was not related to HGS (Lines 223-235).
- line 246 & 250: the reference 22 seems to be wrong in both lines - should be reference 23 in my opinion. Please re-check all references!
Authors: The comments are most appreciated. As you pointed out, reference number 23 was correct. We re-checked all references. The reference number was changed from 23 to 25 as references were added during the revision process (Line 271 & 275).
Reviewer 3 Report
My general comment is that the manuscript is not clear whether the eating habits of Koreans do result in different diet quality scores. If those scores are not different then what is novel about the observations. In addition, how would race justify the present study.
L11. Change “Single” to “single”
L17. Change “of was” to “was”
L47. Authors indicate that the eating habits of Korean are different. The authors need the clarify the manuscript how the scores for aMED and DASH compare to other studies and confirm that eating habits do result in different scores for aMED and DASH. If those scores are not different, then the present study seems to confirm observations in non-Koreans.
In addition, I suggest the authors strengthen the introduction using Jeong et al Associations of recommended food score and physical performance in Korean elderly.BMC Public Health. 2019 Jan 30;19(1):128.
L90. Please clarify “KNHANES conducts a 24-h recall survey for all participants aged over 12 months”
Table 1. Please define MUFA and SFA.
L194. Change “acitivity” to ”activity”
Table 4. The CI need to be consistently reported , e.g. (lower value-higher value).
Table 4. Define “ref” in the legend.
Author Response
Reviewer's Comments
My general comment is that the manuscript is not clear whether the eating habits of Koreans do result in different diet quality scores. If those scores are not different then what is novel about the observations. In addition, how would race justify the present study.
Authors: Thank you for your critical comments and helpful suggestions. Please see our detailed responses below.
L11. Change “Single” to “single”
Authors: Thank you for this suggestion; we have made the correction (Line 11).
L17. Change “of was” to “was”
Authors: Thank you for finding this typo; we have made the correction (Line 17).
L47. Authors indicate that the eating habits of Korean are different. The authors need the clarify the manuscript how the scores for aMED and DASH compare to other studies and confirm that eating habits do result in different scores for aMED and DASH. If those scores are not different, then the present study seems to confirm observations in non-Koreans.
In addition, I suggest the authors strengthen the introduction using Jeong et al Associations of recommended food score and physical performance in Korean elderly. BMC Public Health. 2019 Jan 30;19(1):128.
Authors: The valuable suggestion is most appreciated. As you pointed out, the sentence written in the previous version of manuscript, “However, these results are difficult to apply to Koreans, due to differences in race or eating habits.” is misleading, so we removed it. We revised the content of the Introduction by quoting Jeong et al.’s research (Lines 50-54).
L90. Please clarify “KNHANES conducts a 24-h recall survey for all participants aged over 12 months”
Authors: Thank you for this suggestion. Based on your suggestion, we have modified this sentence to read more clearly (Line 93).
Table 1. Please define MUFA and SFA.
Authors: As suggested, we have defined MUFA and SFA in Table 1.
L194. Change “acitivity” to ”activity”
Authors: Thank you for finding this typo; we have made the correction (Table 4).
Table 4. The CI need to be consistently reported , e.g. (lower value-higher value).
Authors: Thank you for finding this typo; we have made the correction (Table 4).
Table 4. Define “ref” in the legend.
Authors: As suggested, we have defined “ref” in the legend of Table 4.
Round 2
Reviewer 2 Report
The authors answered all my questions, however, after revision additional points, which should be considered
- Line 247-260: Although you added a sentence about diet Quality Indices, the discussion regarding the possible links between diet quality and muscle mass/strength is weak. Please further discuss the relationship between healthy dietary patterns, their link to a healthy Lifestyle, which is further linked to more physical active... -> more exercise -> reduced loss of muscle mass and function...
- Line 264: "...better Lifestyle than those WHO WITH lower..." Correct this sentence.
- Line 265: define or ad values for HIGH in "high BMI"
- Line 266: define or ad values for HIGHER in "higher energy intake" ...HIGHER compared to what?
- Line 268: ad reference values for "high BMI and high energy intake"
Author Response
Response to Reviewer #2
Reviewer's Comments
The authors answered all my questions, however, after revision additional points, which should be considered
Authors: Thank you for your critical comments and helpful suggestions. Please see our detailed responses below.
- Line 247-260: Although you added a sentence about diet Quality Indices, the discussion regarding the possible links between diet quality and muscle mass/strength is weak. Please further discuss the relationship between healthy dietary patterns, their link to a healthy Lifestyle, which is further linked to more physical active... -> more exercise -> reduced loss of muscle mass and function...
Authors: Thank you for your suggestion. Based on your suggestion, we have revised the discussion section (Lines 247-250).
- Line 264: "...better Lifestyle than those WHO WITH lower..." Correct this sentence.
Authors: This has been revised (Line 265).
- Line 265: define or ad values for HIGH in "high BMI"
Authors: This has been revised (Lines 266-267).
- Line 266: define or ad values for HIGHER in "higher energy intake" ...HIGHER compared to what?
Authors: This has been revised (Lines 267-268).
- Line 268: ad reference values for "high BMI and high energy intake"
Authors: We revised the sentence so that there is no misunderstanding in interpretation (Lines 269-270).
Reviewer 3 Report
Ls 128-132. Twice the same sentence. Please delete one.
Author Response
Response to Reviewer #3
Reviewer's Comments
Ls 128-132. Twice the same sentence. Please delete one.
Authors: The critical comments are most appreciated. We removed one of the sentences. (Lines 128-130).